# Building National Open Science Cloud Initiatives (NOSCIs) in Southeast Europe: Supporting Research and Scholarly Communication

**Milica Ševkušić** [1,*], **Eleni Toli** [2,*], **Katerina Lenaki** [3], **Kalliopi Kanavou** [2], **Electra Sifakaki** [2], **Biljana Kosanović** [4], **Ilias Papastamatiou** [5] and **Elli Papadopoulou** [2]

1   Institute of Technical Sciences of SASA (Serbian Academy of Sciences and Arts), 11000 Belgrade, Serbia
2   Athena Research & Innovation Centre, 15125 Marousi, Greece
3   Ministry of Education and Religious Affairs, 15122 Marousi, Greece
4   University of Belgrade Computer Centre, 11000 Belgrade, Serbia
5   National Infrastructures for Research and Technology—GRNET S.A., 11523 Athens, Greece
*   Correspondence: biblioteka@itn.sanu.ac.rs (M.Š.); elto@athenarc.gr (E.T.)

**Abstract:** The Horizon 2020 project National Initiatives for Open Science in Europe—NI4OS Europe supports the development of the European Open Science Cloud (EOSC) by integrating 15 countries in Southeast Europe into the governance structure of this new pan-European research environment. Through a qualitative secondary analysis of the data collected during the project, the paper focuses on the main instrument developed by the project with the aim of enabling the integration of the partner countries in the EOSC—a network of national Open Science Cloud Initiatives (NOSCIs)—and explains how the concept of NOSCI and a wide range of related activities, tools, services, and resources foster research and open scholarly communication. The paper has three main sections: the first identifies challenges to scholarly communication in Southeast Europe, the second describes the methodology used to deal with these challenges revolving around the concept of NOSCI, whereas the third presents a set of indicators to track the change generated by project actions and discusses the impact of this methodology and project outputs in the area of scholarly communication.

**Keywords:** European Open Science Cloud; NI4OS-Europe; National Open Science Cloud Initiatives; Open Science; national policies; Southeast Europe; scholarly communication

## 1. Introduction

National Open Science Cloud Initiatives (NOSCIs) are national-level coalitions of Open Science stakeholders that seek to facilitate the integration of EU Member States and Associated Countries in the European Open Science Cloud (EOSC). Their establishment is associated with the acceleration of Open Science (OS) transformation and consequent scholarly communication optimisation worldwide. The purpose of this paper is to shed light on the contribution of the "National Initiatives for Open Science in Europe—NI4OS Europe" project in building NOSCIs [1] in 15 countries of Southeast Europe (Albania, Armenia, Bosnia and Herzegovina, Bulgaria, Croatia, Cypris, Georgia, Greece, Hungary, Moldova, Montenegro, North Macedonia, Romania, Serbia, Slovenia, https://ni4os.eu/partners/, accessed on 2 September 2022) and its impact on all aspects of scholarly communication.

We consider scholarly communication to be defined as the process in which scientists share views and findings regarding their subject of research, contributing thus to the progress of peer-reviewed scientific knowledge worldwide. Closely linked to the research cycle, it is generally considered to include three distinct stages. According to Graham [2], it is a system that involves (a) the informal communication within scientific



networks—where research and idea flows are generated, (b) the initial public dissemination of research results—such as in preprints and conferences and finally (c) the formal publication of research output so that it is available to scientists and the public—mainly in scientific journals.

Technological developments of the last two decades, especially high capacity, linked networks and advanced electronic tools and services have brought major changes in all stages of scholarly communication and increased expectations for better and more open science. Research workflows nowadays reflect the great variety of sources and means, including distinct yet interdependent phases (discovery, analysis, writing, publication, outreach, assessment) [3], and scientists mostly publish in primary sources online (e.g., in working papers, preprints, reports, theses and dissertations, journals, monographs, conference proceedings and patents according to Das, 2015 [4]). Scholarly communities as a whole, i.e., scientists, universities and their libraries, publishers, government bodies, research councils, funders, as well as readers [2], have a vital impact on the scholarly communication system regarding new alternatives in access, publishing and evaluation of research content. They are responsible for creating the conditions for bibliodiversity, that is, the diversity in services, funding and evaluation mechanisms [5]. Taking this into account, their role is crucial in order to achieve the transition to open access and open science and the "accommodation of the different workflows, languages, publication outputs, and research topics that support the needs and epistemic pluralism of different research communities" [5]. The diversification of the concept of scholarly communication and its high importance is also reflected in the EOSC Portal data model, where Scholarly Communication is a subcategory of Sharing and Discovery and is further divided into Analysis, Assessment, Discovery, Outreach, Preparation, Publication, Writing and Other [6].

The relationship between EOSC and scholarly communication is reciprocal and multidimensional. EOSC should facilitate a shift from the "current state of the art towards an Open Science scholarly communication ecosystem that is based on incentives and facilitates Open Science principles and practices in performing and sharing science" [7]. At the same time, scholarly communication on the EOSC vision and services can support the further widening of OS in Europe, the engagement of research communities in EOSC and the broader use of its services. It is therefore important to highlight even more the existing links between EOSC and scholarly communication, by addressing the whole spectrum of EOSC priorities and aims at all levels (policy, governance, users, technical issues). The complexity of the EOSC universe often does not allow scholarly communities to immediately observe and understand the benefits of EOSC. Their further engagement in the EOSC discourse will certainly enrich the OS discussion in Europe.

As already mentioned, the project National Initiatives for Open Science in Europe—NI4OS Europe works towards widening OS and EOSC in Southeast Europe. It is one of the four EOSC Regional Projects and is part of the INFRAEOSC-05 Collaboration (https://ec.europa.eu/info/funding-tenders/opportunities/portal/screen/opportunities/topic-details/infraeosc-05-2018-2019). The project was launched in September 2019 and will end in February 2023 (more details: https://cordis.europa.eu/project/id/857645; project website: https://ni4os.eu – accessed on 6 September 2022). It supports the development of the EOSC by contributing to its portfolio of services and the establishment of NOSCIs, engaging national and regional research communities of Southeast Europe in the EOSC governance, strengthening Open Science (OS) practices and promoting the FAIR principles [8].

This paper highlights the scholarly communication perspective and demonstrates how approaches, instruments, services and tools developed during the project help to establish sustainable and networked local environments for research and open scholarly communication in terms understandable to its main target audience—the research community. The presented analysis may as well be useful to policy-makers in other regions and especially in developing countries. The region of Southeast Europe is marked by a great diversity of local contexts, due to which the project team had to devise flexible solutions that could

be replicated in various environments. This gives to the project findings (challenges and solutions) a higher value, as wider OS and EOSC communities can benefit from them.

The paper has three main sections: the first describes the data and methodology used to deal with the challenges revolving around the concept of NOSCI in the target area, the second identifies challenges to scholarly communication in Southeast Europe at the regional level, whereas the third presents a set of indicators to track the change generated by project actions and discusses the impact of this methodology and project outputs in the area of scholarly communication.

## 2. Data and Methodology

The paper uses a qualitative secondary analysis to show whether the solutions offered by the project have an impact on scholarly communication. It does not involve original research. The following data collected during the project, using various methods, are reused in this paper:

- Information about Open Science stakeholders in the region, collected during the NI4OS-Europe landscaping activity (expert survey) [9];
- Country sheets analysis: report from the EOSC Executive Board Working Group (WG) Landscape (expert survey) [10];
- Survey data collected in the autumn of 2019 for the NI4OS-Europe landscaping study (survey) [11];
- Analysis of Open Science policies in the region and across Europe (desk analysis) [1];
- Project partners' reports about NOSCI development (expert survey).

Publicly available statistical data provided by EUROSTAT and the World Bank are also used.

Based on these sources of information, and primarily the stakeholder information and the survey data, direct and indirect challenges to scholarly communication were identified. Desk analysis is used to explain the project's response to these challenges provided through the overarching concept of NOSCI. The analysis is entirely focused on the regional level and is limited to the challenges that are specific to the region of Southeast Europe and those that are shared by the countries involved in the project. Challenges present in individual countries go beyond the scope of this paper. The reason for this lies in the fact that the project has not dealt with each country as an isolated case study but has rather sought to devise more general solutions that are locally applicable.

To measure the impact of the response, we introduce a set of indicators derived from the identified challenges. In the process of defining the indicators for this study, the indicators for monitoring EOSC readiness [12] and NOSCI establishment [1,13] were analysed. Keeping in mind that these indicators are intended for monitoring at the national level and do not always apply to scholarly communication, it was necessary to select those that are relevant for scholarly communication and adjust them to regional-level monitoring. The mapping between the indicators defined for this study and the NI4OS-Europe Blueprint metrics [13] and EOSC readiness indicators [12] is shown in Table 5.

The analysis follows the challenge—response—impact matrix, which is reflected in the structure of the paper.

## 3. Landscaping

### 3.1. Remarks on the Landscaping Survey and Regional Challenges

The challenges to scholarly communication in the NI4OS-Europe partner countries were identified at the regional level and are based on the data collected during the landscaping activity at the beginning of the project. The data include the literature data, information provided by national experts, and, most importantly, the data collected in a survey conducted in the autumn of 2019, which also provided an input to the EOSC Secretariat's Landscape Activity The five INFRAEOSC-5b projects conducted landscaping activities in a coordinated manner and their inputs were eventually aggregated and analysed in a study commissioned by the EOSC Secretariat [14]. This initial mapping of the existing

Open Science (OS) initiatives, infrastructures, services, policies, stakeholders and topics in each of the partner countries at the beginning of the project [11], which helped to shape further actions, serves as the starting point for the identification of challenges to scholarly communication in our analysis. The relevance of information collected on this occasion for the present analysis is best illustrated by the fact that 61.2% (30 out of 49) of the questions in the landscaping questionnaire for stakeholders performing research. The survey included five questionnaires—one for each stakeholder group. In this paper, we are focusing on the main actors of scholarly communications—those who perform research. More information about the stakeholder groups and the structure of the survey: [11] were directly related to scholarly communication. Table 1 summarises the topics covered by the questionnaire and indicates their relation to scholarly communication.

**Table 1.** Topics covered in the questionnaire for research performing stakeholders: relation to scholarly communication.

| Main Topic | Specific Topics | Relation to Scholarly Communication | No. of Questions |
|---|---|---|---|
| General | | | 4 + 1 |
| Profile of the organization | domain and size | | 3 |
| | content and rights | Direct/Publication | 1 |
| Funding | criteria | Indirect | 2 |
| | user support (for services) | Indirect | 1 |
| Policies | institutional | Direct/Discovery, Outreach, Publication, Assessment | 1 |
| | infrastructure roadmap | Indirect | 1 + 1 |
| | OS compliance | Direct/Discovery, Outreach, Publication, Assessment | 1 |
| | research assessment | Direct/Assessment | 1 + 2 |
| Infrastructure | for OS | Direct/Discovery, Outreach, Publication | 1 |
| | needs and preferences | Indirect | 1 |
| Services | needs and preferences | Indirect | 1 |
| Open Science | training | Direct/Skills and competencies | 3 |
| | publication repositories | Direct/Discovery, Outreach, Publication | 1 + 6 |
| | data repositories | Direct/Discovery, Outreach, Publication | 1 + 7 |
| FAIR | awareness | Direct/Analysis, Preparation, Writing, Discovery, Outreach, Publication | 1 |
| | implementation | Direct/Analysis, Preparation, Writing, Discovery, Outreach, Publication | 3 |
| | support | Direct/Skills and competences | 1 |
| EOSC | awareness | Indirect | 2 + 2 |

The "direct" relation means that a question refers to one or more stages of scholarly communication, policies, infrastructure, services, skills and competencies for scholarly communication, or the assessment of research outputs. For example, the investment in a FAIR-compliant repository creates a channel for the dissemination of research outputs. The subcategories of scholarly communication from the EOSC portal [6] are used in the table to describe more precisely the direct relation. The "indirect" relation means that the subject of a question has an indirect impact on scholarly communication, e.g., the main purpose of the user support for services is not necessarily to improve scholarly communication; however, it may not only have such an effect, but it is very likely that researchers will see it as being in the service of scholarly communication. This "scholcomm-centric" perspective among researchers is reflected in the responses to the question "What do you expect from EOSC?", which predominantly referred to concepts directly related to scholarly communication (Table 2) This open-ended question was answered by less than 40% of the respondents. The respondents could mention as many concepts as they wished, and some even provided

descriptive answers. Their answers were analysed to extract distinct concepts, which were further normalised. Only the concepts mentioned more than five times are shown in the table.

**Table 2.** Researchers' expectations from EOSC: survey results.

| Concept | Mentions |
|---|---|
| access | 62 |
| data | 44 |
| services | 27 |
| support | 16 |
| training | 15 |
| computing resources and services | 15 |
| sharing | 13 |
| collaboration | 10 |
| Open Science | 10 |
| research outputs | 10 |
| Open Access | 9 |
| repositories | 9 |
| information | 8 |
| infrastructure | 7 |
| integration | 6 |
| networking | 6 |
| publications | 6 |
| resources | 6 |
| guidance | 5 |
| Open Data | 5 |
| research support | 5 |
| standardisation | 5 |

Southeast Europe is a highly diversified region in political, social and economic terms, which determine the research environment and also have an indirect impact on scholarly communication.

Accordingly, the NI4OS-Europe project team had to take into account some overall challenges related to this broader social and political context. Over the past three decades, the region has witnessed turbulent changes. Out of the 15 partner countries, 9 were part of 2 federations that no longer exist—USSR (Armenia, Georgia, Moldova) and the former Yugoslavia (Bosnia and Herzegovina, Croatia, Montenegro, North Macedonia, Serbia, Slovenia). Eight countries (Albania, Bulgaria, Croatia, Hungary, Montenegro, North Macedonia, Romania, Slovenia) became members of the North Atlantic Treaty Organisation (NATO), four of them (Albania, Bulgaria, Hungary, Romania) had previously belonged to the opposing military alliance (Treaty of Warsaw). Thirteen partner countries have experienced major political transformations as a result of the collapse of communism. Three decades ago, only Greece was a member of the European Union. In the meantime, the EU has been extended to include six more partner countries (Bulgaria, Croatia, Cyprus, Hungary, Romania, Slovenia) [15]. Finally, the political and military conflicts that accompanied the break-up of Yugoslavia disrupted the research environment in the Balkans and it took 15 years to recover research collaboration to the level before the conflicts [16]. This broader socio-political and historical context will not be discussed further, but it has to be taken into consideration, as it largely explains infrastructural fragmentation and the lack of collaboration towards developing, for example, regionally coordinated publishing services or aggregators.

In the following sections we provide some additional insights on the challenges identified through our landscaping activity that are of particular interest for scholarly communities.

### 3.2. The Place of Open Science in the Research Governance Systems

Among other things, the success of initiatives and policies is related to the position of the authority or body behind them to impose relevant regulations and mobilise various stakeholders and research communities. In most countries in the region, regulations relating to OS were adopted by ministries responsible for education, science and research. OS may also be included in a national digital agenda. In Hungary, the Ministry of Innovation and Technology is responsible for OS, while in Georgia, OS policy development is coordinated by the Shota Rustaveli National Science Foundation. Since 2019, research activities in Greece are under the responsibility of the Ministry of Development and Investment while at the same time a general framework for open science is included in the Digital Transformation Strategy 2020–2025 of the Ministry of Digital Governance. It is noteworthy that in Cyprus OS policy-related activities are supported by OpenAIRE, whereas in Armenia the policy is developed within an Erasmus+ project [8].

The question inevitably arises whether individual ministries or funders are well-positioned to cover all relevant topics in policies and ensure their wide implementation. This is also a challenge for scholarly communication because policies and funding relating to Open Science developed by a single ministry may not address all relevant aspects of scholarly communication. On the other hand, involving all relevant ministries, funders and stakeholders in the policy development and implementation process may be difficult to coordinate.

### 3.3. Different Policy Traditions

Although the landscaping survey addressed various levels (national, institutional) and aspects of policies (open access to publications and research data, sharing software under free licences, the preservation of scientific information, information and data security, rules regarding repositories, publishing platforms, FAIR principles, intellectual property rights, access to services and terms of use), the responses were inconsistent, especially as regards institutional policies [11]. Additional efforts were made to refine the data through an expert survey [1,8].

In the autumn of 2019, most countries did not have a national OS policy. The existing policies mostly addressed publications. Some also mentioned research data, but very few discussed the FAIR dimension of them. This suggests a rather conservative perception of scholarly communication as being limited to communicating research through publications only.

In most countries it was possible to find some documents addressing OS explicitly or individual aspects of OS implicitly. These documents took various formats (platform, strategy, agenda, plan) and approaches (mandating vs. recommending), while the documents were adopted by different bodies. Due to this, devising a single model for the alignment of policies with OS principles would be ineffective.

### 3.4. Non-EU Countries Are Less Integrated in European Open Science Networks

Throughout the NI4OS-Europe project, participation of the partner countries in the so-called "EOSC pillars" (consortia, associations or networks contributing to the development of EOSC) has been monitored as an indicator of progress towards EOSC integration. The list of the "pillars" is as follows: OpenAIRE (OpenAIRE—Open Access Infrastructure for Research in Europe) is a European infrastructure for Open Science: https://www.openaire.eu, accessed on 20 July 2022); European Grid Infrastructure (EGI, which seeks to provide access to high-throughput computing resources in Europe using grid computing techniques: https://www.egi.eu, accessed on 20 July 2022); Research Data Alliance (RDA, a research community organisation striving to facilitate open data sharing at a global level, founded in 2013 by the European Commission, the American National Science Foundation and National Institute of Standards and Technology, and the Australian Department of Innovation; it relies on a network of national RDA nodes: https://www.rd-alliance.org/, accessed on 20 July 2022); and GÉANT (Gigabit Research and Education Network, which is

a pan-European network connecting national research and education networks (NRENs) across Europe: https://geant.org/, accessed on 20 July 2022). The list has been limited to the "ones that are relevant for the majority of the NI4OS-Europe countries" [1]. In line with the focus of this study, we will use the term "Open Science networks" and extend the list to include a number of organisations, initiatives or European Research Infrastructure Consortia (ERICs) involved in EOSC-related projects and the development of infrastructure, services, tools, guidelines and best practice relevant for open scholarly communication. These are EUDAT (European Collaborative Data Infrastructure, which is a European infrastructure that integrates data services and resources supporting research: https://www.eudat.eu/ accessed on 20 July 2022); National Research Infrastructure Roadmap (https://www.esfri.eu/national-roadmaps, accessed on 20 July 2022); CESSDA (Consortium of European Social Science Data Archives, which is a European Research Infrastructure Consortium that offers data services to the social sciences by bringing together social science data archives across Europe: https://www.cessda.eu/, accessed on 20 July 2022); DARIAH (Digital Research Infrastructure for the Arts and Humanities, which is a European Research Infrastructure Consortium that supports digitally-enabled research and teaching across the arts and humanities: https://www.dariah.eu/, accessed on 20 July 2022); EIFL (Electronic Information for Libraries, which is a not-for-profit organization supporting libraries in developing and transition economy countries to gain access to knowledge: https://www.eifl.net/, accessed on 20 July 2022); CLARIN (Common LAnguage Resources and Technology INfrastructure, which is an ERIC offering language data, language technology data processing and expertise to the research community: https://www.clarin.eu/, accessed on 20 July 2022); and OPERAS (a European Research Infrastructure dedicated to open scholarly communication in the social sciences and humanities: https://www.operas-eu.org/ accessed on 20 July 2022). Involvement in such networks offers to participants various opportunities (e.g., to participate in projects and training; qualify for technical support; have access to infrastructure, tools and services, data and guidelines; and exchange information, etc.).

Figure 1 shows the NI4OS-Europe partner countries represented in each network at the beginning of the project, in the autumn of 2019 (light grey indicates pending initiatives to join a particular network. In all four cases, the initiatives were successful). It is apparent that the non-EU countries are considerably less involved in the activities of the selected networks. We did not investigate the reasons for this, and we can merely speculate that they may include the lack of familiarity with the networks, the lack of interest among relevant stakeholders at the national level, the lack of funds to pay participation fees (where required), etc.

| | Country | Open AIRE | EUDAT | EGI | GEANT | National RI Roadmaps | RDA NODE | CESSDA | DARIAH | EIFL | CLARIN | OPERAS |
|---|---|---|---|---|---|---|---|---|---|---|---|---|
| EU | Bulgaria | ■ | | ■ | ■ | ■ | | ■ | | | ■ | |
| | Croatia | ■ | | ■ | ■ | ■ | | ■ | | | ■ | ■ |
| | Cyprus | ■ | ■ | | ■ | | | ■ | | | ■ | |
| | Greece | ■ | ■ | ■ | ■ | ■ | ■ | ■ | ■ | | ■ | ■ |
| | Hungary | ■ | ■ | ■ | ■ | ■ | | ■ | ■ | | ■ | |
| | Romania | ■ | ■ | ■ | ■ | ■ | ■ | | ■ | | ■ | |
| | Slovenia | ■ | ■ | ■ | ■ | ■ | ■ | ■ | ■ | ■ | ■ | ■ |
| Non-EU | Albania | | | | ▨ | | | | | ▨ | | |
| | Armenia | ▨ | | | ▨ | | | | | | | |
| | Bosnia & Herzegovina | ░ | | | | | | ▨ | | | | |
| | Georgia | | | | ▨ | | | | | | | |
| | Montenegro | ░ | | | | | | ▨ | | | | |
| | Moldova | | | | ▨ | | | | | ▨ | | |
| | North Macedonia | ░ | | ▨ | ▨ | | | | | | | |
| | Serbia | ▨ | | | ▨ | | | ▨ | ▨ | ▨ | | ░ |

**Figure 1.** Participation of the NI4OS-Europe partner countries in European Open Science networks.

Poor involvement with international OS networks bears risks, such as infrastructure obsolescence or investment in unsustainable tools and services that are not compliant with standards. This may have negative effects on scholarly communication, especially in the long term. Local stakeholders may stay behind major developments in this area, such as the development of Open Access book publishing platforms, efforts to create multilingual controlled vocabularies, discussions about research evaluation, etc.

*3.5. Varying Structure of OS Stakeholders across Countries*

The mapping of OS stakeholders in the partner countries was part of the landscaping action at the beginning of the project [11]. The data were provided by the project partners after the stakeholder groups had been defined. This was done in two steps: the project partners provided either a preliminary list of the stakeholders or just the numbers per group; the project partners were incited to revise the preliminary lists or to provide a list with contact information (if they had previously provided only numbers). The stakeholder map was made based on the data collected in the second step. The main purpose of this action was identifying institutions, infrastructures and services to be targeted by project activities and, in particular, providing input for the model of national OS initiatives and the EOSC Landscape Activity [14].

Five stakeholder groups were defined:

1. Funders and policymakers—FUND (the actors who fund research and, most commonly, shape research-related policies;
2. Those who perform research—CREATE (research performing organisations and researchers);
3. Those who perform research—SUPPORT (repositories, research infrastructures, e-infrastructures, service providers, libraries);
4. Those who "consume" research—CONSUME (SMEs and citizens);
5. OS facilitators (including OS initiatives)—FACILITATE (international nodes, coordinators).

Table 3 shows the number of stakeholders per group identified in the partner countries. The completeness and reliability of the data may be disputed, as it is possible that the project partners were more familiar with a particular group of stakeholders or failed to identify others. In addition, there are some differences between the initial input and the final map, e.g., there are no members of the FUND group for Croatia in the final map, whereas during the initial data collection there were two. Similarly, there are no SUPPORT stakeholders for Montenegro in the final map, whereas there were 10 during the initial data collection. We assume that the data in the map are more accurate because during the initial data collection some partners provided just numbers. However, all project partners are organisations that are expected to have a good insight into the local situation and it is reasonable to assume that the data provided by them do not significantly deviate from the actual situation.

It is interesting that Cyprus and Greece have considerably more research funders and policy makers than the other countries, which may suggest less centralised research systems. Slovenia has the greatest number of EOSC facilitators (EOSC Working Group representatives, CESSDA representatives, etc.), which reflects its presence in various European networks. Apart from researchers, the SUPPORT group is the most directly relevant for scholarly communication, as it includes stakeholders who provide and maintain infrastructure (repositories, publishing platforms, e-infrastructure) and provide relevant services (metadata creation, PID assignment, dissemination, etc.), as well as training. If this group is insufficiently represented in a country, there is a risk that infrastructure development and skills will lag even if relevant policies are in place and funding is sufficient.

**Table 3.** Open Science stakeholders in the NI4OS-Europe partner countries.

| Country | Fund | Create | Support | Consume | Facilitate |
|---|---|---|---|---|---|
| Albania | 2 | 14 | 3 | 0 | 0 |
| Armenia | 4 | 23 | 0 | 1 | 0 |
| Bosnia & Herzegovina | 3 | 11 | 10 | 6 | 1 |
| Bulgaria | 3 | 17 | 8 | 5 | 3 |
| Croatia | 0 | 54 | 15 | 3 | 0 |
| Cyprus | 20 | 32 | 13 | 7 | 0 |
| Georgia | 1 | 14 | 4 | 1 | 0 |
| Greece | 15 | 58 | 48 | 11 | 8 |
| Hungary | 6 | 52 | 28 | 4 | 8 |
| Moldova | 3 | 46 | 30 | 2 | 5 |
| Montenegro | 1 | 5 | 0 | 6 | 0 |
| N. Macedonia | 4 | 21 | 4 | 3 | 0 |
| Romania | 6 | 27 | 33 | 24 | 2 |
| Serbia | 4 | 106 | 17 | 10 | 3 |
| Slovenia | 6 | 63 | 43 | 4 | 16 |

*3.6. Varying Levels of Available Funding*

Investment in research development in the NI4OS-Europe partner countries is generally lower than in the rest of Europe, and this is primarily due to their lower economic performance. While most EU members among the partner countries belong to High Income Economies according to the World Bank ranking [17], the associated countries mostly fall into the group of Upper-Middle-Income Economies. Table 4 presents the research and development expenditure in the countries in the region, which is considerably lower than in the EU: in all partner countries it is below the EU average (2.2% of GDP) and only in Slovenia is it close to the EU average [18]. EU members among the partner countries were more actively involved in Horizon 2020 projects and they, accordingly, received more funds [19].

**Table 4.** NI4OS-Europe partner countries in 2019: World Bank Country Ranking, research and development expenditure (percent of GDP) and the net contribution in Horizon 2020.

| Category | Country | World Bank Country Ranking Income | Research and Development Expenditure (% of GDP) | Net EU Contributionin Horizon 2020 (Million EUR) |
|---|---|---|---|---|
| EU members | Bulgaria | UM | 0.83236998 | 161.25 |
| | Croatia | H | 1.08105004 | 137.97 |
| | Cyprus | H | 0.71463001 | 319.47 |
| | Greece | H | 1.27566004 | 1710.82 |
| | Hungary | H | 1.47736001 | 371.13 |
| | Romania | H | 0.47832 | 300.69 |
| | Slovenia | H | 2.04703999 | 379.85 |
| Non-EU members | Albania | UM | no data | 5.79 |
| | Armenia | UM | 0.17854001 | 4.20 |
| | Georgia | UM | 0.28468001 | 8.71 |
| | Moldova | LM | 0.23672 | 7.36 |
| | Bosnia & Herzegovina | UM | 0.19264001 | 8.72 |
| | Montenegro | UM | 0.36328 | 4.62 |
| | N. Macedonia | UM | 0.36783001 | 14.80 |
| | Serbia | UM | 0.88666999 | 134.85 |

Sources: World Bank; Horizon Dashboard. LM—Low-middle-income. UM—Upper-middle-income. H—High income. GDP—Gross Domestic Product. EU—European Union.

The size and structure of funding for scholarly communication in individual countries are difficult to estimate. This topic was beyond the scope of the NI4OS-Europe landscaping

activity. On the qualitative level, it is important to mention that the survey data reveal that in all countries in the region publicly funded institutions provide funding and technical support for repositories and journal publishing platforms. In Bulgaria, Croatia, Romania, Serbia and Slovenia, national funders provide subsidies for scholarly journals, but only in Croatia and Slovenia is support provided only to Open Access journals [20].

### 3.7. The Lack of Information across the Region

At the beginning of the project, there were no curated national or regional registries offering information about relevant institutional stakeholders, nor were there any standardised and curated national or regional registries of services relevant for OS. The information provided by international registries, such as OpenDOAR, re3data, FAIRSharing, etc. was incomplete. The most reliable source of information about OS policies were the OpenAIRE country pages, for the countries involved in OpenAIRE. Due to this, it was impossible to verify the information about infrastructure and policies collected in the landscaping survey [11].

The lack of information is also a serious challenge for scholarly communication. It makes it difficult to identify potential partners, which may discourage collaboration. Additionally, there is a risk of effort duplication, instead of replicating or sharing sustainable solutions developed within the region.

### 3.8. The Lack of Incentives for Open Science

Two sources of information about incentives for OS were used: the landscaping survey data (questions about institutions' internal rules and the aspects taken into account when evaluating researchers) and an expert survey presenting four case studies—Croatia, Greece, Serbia and Slovenia [8].

The responses to the question about internal rules for OS-related topics (intellectual property rights, open-source software, publishing platforms, article and processing charges) suggest that in most institutions across the region, OS practices were encouraged. However, the data are inconsistent with the findings of the expert survey and we will not analyse them further. As far as research evaluation is concerned, the survey data show that publications were identified as the most important parameter for researcher evaluation in the NI4OS-Europe partner countries. On the other hand, data sharing, software, OS and OA, social outreach, knowledge transfer, and citizen science were recognised as little to moderately important. The radar diagram of Figure 2 is based on the responses to the question "Which of these aspects are taken into account most when evaluating researchers?" provided by researchers and representatives of research performing organisations (CREATE). The respondents were expected to assess 14 options (13 options presented in the diagram and "Other") as not important, little important, moderately important, important, or very important. The responses were translated into numerical values and an average value was calculated.

In brief, throughout the region, there were hardly any incentives for Open Science activities beyond policy encouragement. This lack of incentives did not encourage innovation in scholarly communication and it can be argued that it undermined the development and sustainability of emerging platforms and initiatives.

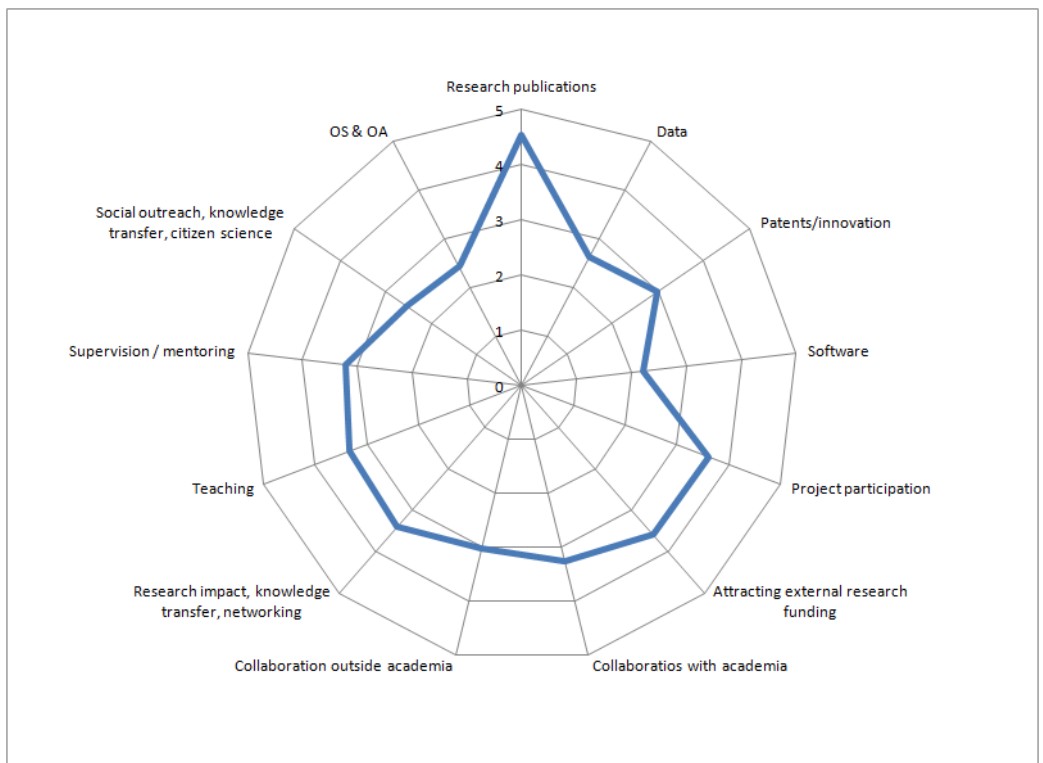

**Figure 2.** The importance of various parameters in research evaluation (survey data).

*3.9. Uneven Infrastructure Development*

The project did not deal specifically with the infrastructure for scholarly communication. It covered generic (cloud computing, data archiving and discovery services, etc.) and thematic services, as well as repositories [21]. In line with the inclusive definition of scholarly communication, all these services support various phases of scholarly communication (discovery, analysis, and partly publication). However, publishing platforms and services for writing, outreach, and assessment were beyond the scope of the project, though some of them (journal publishing platforms and e-learning resources) were captured in the landscaping analysis. As the comprehensive service catalogues are yet to be developed and the survey data are often inconsistent—which may reflect poor awareness and a lack of interest—we do not have sufficient data for a detailed analysis of the infrastructure in the region. Still, some immediately observable facts clearly demonstrate that access to the infrastructure relevant for scholarly communication varies across countries. Although it is apparent that, for example, thematic services for archaeology and heritage research are more developed in Greece and Cyprus than in other countries in the region, we will not try to make any conclusions based on the distribution of thematic services mainly because it was impossible to assess their availability to various research communities in a country and, accordingly, their impact.

The NI4OS-Europe partner countries that are members of the European Union and are involved in various European networks have better access to shared, international infrastructures than the Associated Countries, though this gap has been mitigated through projects aiming to establish pan-European infrastructures (Grid computing infrastructure, OpenAIRE services, etc.) [1].

The integration of repositories with OpenAIRE could be another indicator of infrastructure development. In order to be harvested by OpenAIRE, repositories have to meet certain technical requirements. A small number of repositories from a country may suggest either that there are no repositories, that the existing repositories are not interoperable, or that there is a lack of awareness about the importance of interoperability with infrastructures, such as OpenAIRE. Croatia and Serbia had the largest number of publication repositories

integrated with OpenAIRE. At the same time, there were five countries (Albania, Bosnia and Herzegovina, Georgia, Montenegro, and Romania) that had no repositories harvested by OpenAIRE. It is noteworthy that Croatia had the greatest number of repositories thanks to a publicly funded national repository infrastructure [22].

In the landscaping analysis, data repositories were discussed separately from publication repositories. A small number of data repositories were identified in the survey (14), five of which were in Greece and three were in Slovenia. The data repository registry re3data (https://www.re3data.org/, accessed on 26 July 2022) listed data repositories from Bosnia and Herzegovina, Croatia, Greece, Hungary, Romania and Slovenia. However, one of the repositories was incorrectly associated with Bosnia and Herzegovina, instead of Serbia [11].

Poorly developed infrastructure can not only limit the visibility of local research but can also discourage research communities from adopting open practices.

### 3.10. Undeveloped Training for Open Science

The question about the training and support provided by institutions was strongly focused on various aspects of scholarly communication (repositories, research data, publishing platforms, persistent identifiers, licences, article and book processing charges, intellectual property rights, open-source software, open educational resources and open practices). The survey results have so far been analysed in various contexts using the full range of data for all countries and all stakeholder groups [8,23]. It was concluded that institutions in the NI4OS-Europe partner countries mostly provided training in intellectual property rights and copyright (47%) and repositories (40%), while only 26% provided training in research data management—publishing of open data, FAIR, RDM plans, data protection, data curation, and long-term preservation, as shown in Figure 3. According to the survey data, 22–38% (depending on the topic) of the respondents did not plan to provide support or training [8]. The number of responses collected per country varied and doing the analysis for individual countries did not make sense.

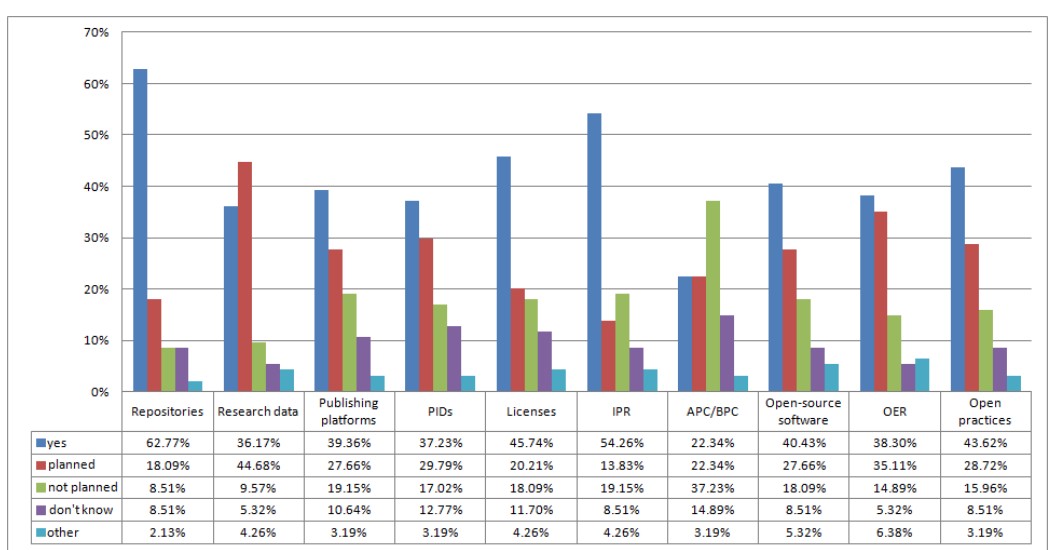

| | Repositories | Research data | Publishing platforms | PIDs | Licenses | IPR | APC/BPC | Open-source software | OER | Open practices |
|---|---|---|---|---|---|---|---|---|---|---|
| yes | 62.77% | 36.17% | 39.36% | 37.23% | 45.74% | 54.26% | 22.34% | 40.43% | 38.30% | 43.62% |
| planned | 18.09% | 44.68% | 27.66% | 29.79% | 20.21% | 13.83% | 22.34% | 27.66% | 35.11% | 28.72% |
| not planned | 8.51% | 9.57% | 19.15% | 17.02% | 18.09% | 19.15% | 37.23% | 18.09% | 14.89% | 15.96% |
| don't know | 8.51% | 5.32% | 10.64% | 12.77% | 11.70% | 8.51% | 14.89% | 8.51% | 5.32% | 8.51% |
| other | 2.13% | 4.26% | 3.19% | 3.19% | 4.26% | 4.26% | 3.19% | 5.32% | 6.38% | 3.19% |

**Figure 3.** Training and support provided by libraries, e-infrastructures, research infrastructures and service providers (survey data).

If we limit the analysis to the SUPPORT group, which includes stakeholders whose mission is related to support and training (libraries, e-infrastructures, research infrastructures and service providers), the share of those who did not plan to organise training is somewhat smaller (8–37%). However, the percentage of those who did not know whether their institutions offered training was not insignificant (5–13%). It is also alarming that for most topics less than half of the respondents from the SUPPORT group provided training.

Unfortunately, we may only speculate about the reasons for the fairly low training and support offer because this issue was not covered by the survey.

### 3.11. Linguistic Diversity

More than 10 different languages, none of which is considered a major European language, are spoken in the NI4OS-Europe partner countries. Five different alphabets are used, which made it difficult to find or identify information in the local languages on institutional websites and analyse policies and services.

In most countries in the region, English is not so widely used in research communities as in other regions of Europe. In practice, this means that the efficiency of training provided in English may be limited. The reusability of materials in local languages is also limited and can be achieved only in some clusters (Greece and Cyprus, Romania and Moldova, among Slavic-speaking partner countries). This is also a challenge in scholarly communication, especially in those disciplines where local languages are predominantly used.

## 4. National Open Science Cloud Initiatives

EOSC constitutes a major ambition in the European Open Science policy, being a federated ecosystem of research infrastructures, e-infrastructures and services that allow the scientific community to share and process publicly funded research results and data across borders and scientific domains. Having the researchers in its core, early enough it was understood that efficient ways needed to be found to engage with researchers and scholarly communities, and clearly convey the message of the whole new possibilities provided by EOSC initiative to the production of research and innovation. Within its scope and ambitions, the EOSC reinforces Open Science, Open Innovation and Open to the world policies and fosters best practices of global data findability and accessibility (FAIR data); helps researchers to get their data skills recognised and rewarded; helps to address issues of access and copyright (IPR) and data subject privacy; allows easier replicability of results and limit data wastage; and it contributes to clarification of the funding model for data generation and preservation, reducing rent-seeking and priming the market for innovative research services.

EOSC stakeholders and related projects had to answer the question of how to promote and support the implementation of the above not only at an overall European level but within their countries and their national research ecosystems. For NI4OS-Europe, an additional difficulty has been posed by the region itself. Acting in an area with high diversity and various OS maturity levels, the implementation of the above cannot take place in a homogenised way. Out of this need, the concept of the National Open Science Cloud Initiatives (NOSCIs) has been developed in response to the specific traits and challenges in the targeted region, based on complex and multilayered analyses of stakeholders, policies and local contexts [1]. NOSCIs can be considered as a coalition of national organisations that have a prominent role and interest in the EOSC and have as their main aim the promotion of synergies at the national level, and the optimisation/articulation of their participation in European and global challenges in this field of Open Science. Similar to scholarly communication, NOSCIs are inclusive and require the involvement of stakeholders from across the research lifecycle. Connecting them at the national level provides not only a testbed for the formulation of OS policies but also a forum for knowledge dissemination and sharing.

To support the establishment of the NOSCIs, the NI4OS-Europe project proposed the Blueprint [1]. This is a holistic framework, inspired by Open Science models and guidelines. It includes modular workflows, a set of indicators and operational aspects for facilitating the establishment, governance and operation of the national initiatives. It adopts an agile approach, as national initiatives can have different formats of organisation and levels of maturity—the NI4OS-Europe Blueprint can be seen as a general "best-case scenario" guideline that gives to countries or to national initiatives maximum flexibility, while making sure that all aspects important to them are addressed.

As part of the Blueprint, a simplified five-step methodology for the establishment of the NOSCIs was introduced, presented in Figure 4. For the purposes of this paper, it may be interesting to highlight its structured yet inclusive approach, which proportionally addresses several aspects of the research workflow phases in scholarly communication: discovery, analysis, writing, publication, outreach and assessment.

**Blueprint**
**Workflows | The five-steps methodology**

Communicate/engage with relevant government officials

Put effort that NOSCI becomes the mandated EOSC organization

1 2 3 4 5

Invite Stakeholders, Start the discussions

Discuss how the NOSCI will operate and draft the MoU + OS National Policies

Spread the know-how. Share knowledge within and outside the consortium

**Figure 4.** A blueprint of the five-step set-up methodology for the NOSCIs.

As in any solid start of an endeavour of this size involving various actors with diverse capacities, an essential first step is to identify local EOSC & OS stakeholders and their roles and in turn design and establish proper communication workflows between them. A landscape review on available (e-)infrastructures and training should follow to create a deep understanding of the current status and bring all stakeholders up to date and on the same path. Communication among different parties is important, therefore regular meetings should be planned. To ensure the sustainability of the whole endeavour, communication and engagement of relevant government officials are crucial elements, which should be confirmed right from the beginning. Last but not least, reaching out to the wider public to communicate the status and goals of Open Science Cloud in the country; informing the public of EOSC updates will advance synergies at the national level and strengthen links to the EOSC. Organising events targeted at the wider public will introduce the NOSCI (even if still under formation) to all Open Science Cloud-related national communities (users, developers, infrastructure providers, funding agencies, related public bodies, industry, etc.).

In addition to the organisational, operational and governance aspects, the Blueprint stresses the important role that policies play in the sharing and promotion of research outputs. In fact, it is important that OS policies support a free flow of knowledge and data and overall access via the Internet, starting from the three main outputs of research: literature, data and software. This approach to the various research outputs, together with the way infrastructures and services are offered and the framework on research assessment and capacity building through skills and training, are or should be important elements in any national OS strategy.

**5. Impact**

The various landscaping activities in relation to EOSC have revealed the different levels of Open Science and EOSC readiness across European Countries. These efforts have been carried out at two levels, by the INFRAEOSC-5 projects through the dedicated thematic task force and by the Landscaping Task Force of the EOSC Executive Board (which was part of the previous governance phase of EOSC before the establishment of the Association. Landscape Working Group | EOSCSecretariat. Retrieved 22 October 2020, from https://www.eoscsecretariat.eu/working-groups/landscape-working-group).

The landscaping has been one of the very first activities for the projects and significantly contributed to the creation of strong collaborative links among them. The outcomes of the landscaping activities soon sparked a discussion among EOSC stakeholders and EOSC supporting projects about the necessity of having an accurate understanding of the status of EOSC readiness in each country and about the methodology and steps that are needed to measure it. This is important not only to understand the starting point for setting up national initiatives supporting EOSC but also to monitor within each Member State both the progress of EOSC as a whole, as well as particular aspects of it, with scholarly communication being one of them.

It is in this frame that the NI4OS-Europe project presented its Blueprint, including an indicative set of metrics that may be used for the assessment of the status and progress of the NOSCI in the region, which is in line with the EOSC Readiness Indicators identified by the former Landscaping Task Force. The NI4OS-Europe indicators have not been created to specifically measure aspects of scholarly communication but rather have a broader scope and can be predominantly used as a guide to complement the establishment and operation of a NOSCI.

Building on this previous work on indicators, the current paper delivers a contribution in relation to scholarly communication at three levels. First, it highlights a subset of indicators of interest for measuring activities in scholarly communication. They are derived from the NI4OS Blueprint metrics and are listed in the "Indicator" column in Table 5. Then, it does a parallel mapping between the scholcomm-related indicators and the NI4OS-Europe Blueprint and the EOSC Readiness Indicators. Finally, it establishes a relation between the proposed indicators relevant for scholarly communication and a number of OS-related challenges identified in the region. To the best of our knowledge, Table 5 provides the first attempt to map and adjust existing indicators in relation to challenges in scholarly communication. In the creation of the blueprint metrics, their reusability potential has been a major criterion from the outset. Now, this selection process in relation to scholarly communication further increases their reproducibility.

**Table 5.** The challenges to scholarly communication in the NI4OS-Europe countries and the mapping between the impact indicators for scholarly communication and the NI4OS-Europe Blueprint metrics and EOSC readiness indicators.

| Challenge | Indicator | Mapping with NI4OS-Europe Blueprint Metrics | Mapping with EOSC Readiness Indicators |
|---|---|---|---|
| Different position of OS in various research governance systems | NOSCIs established | 1.a.1. NOSCI established<br>1.b.1. Form of organisation | National initiative in place/planned |
| Different policy traditions | Open Science policies are in place | 1.d.3. National/Institutional policies around Open Science | OS/FAIR policies supported/ monitored/ planned |
| Non-EU countries are less integrated in European OS networks | Membership in EOSC and major European OS networks | 4b. Metrics concerning membership in international bodies/fora:<br>4.b.1. EOSC Association participation<br>4.b.2. EOSC pillars participation<br>4.b.3.Other | Integration in the EU bodies |
| Varying structure of OS stakeholders across countries | Multistakeholder representation in the NOSCIs | 1.c.1 Membership: number of organizations<br>1.c.2 Membership: type of organisations | National initiative in place/planned<br><br>Stakeholders involved |

**Table 5.** *Cont.*

| Challenge | Indicator | Mapping with NI4OS-Europe Blueprint Metrics | Mapping with EOSC Readiness Indicators |
|---|---|---|---|
| Varying levels of available funding | Funding for scholcomm infrastructure and activities is addressed in OS policies | 4.a.1. National Fund for OS/OSC in place/ planned <br> 4.a.2. Funding: national OSC project <br> 4.a.3. Funding: International and European OSC Projects | National initiative in place/planned Funding <br><br> Specific funding programmes |
| Lack of information across the region | New sources of information created | 2.c.3. National Open Science portal | N/A |
| Lack of incentives for Open Science | Incentives for OS are included in policies | 1.d.3. National/Institutional policies around Open Science | OS/FAIR policies supported/monitored/ planned—Incentives |
| Uneven infrastructure development | Improved infrastructure <br><br> Improved access to infrastructure <br><br> Regional registry | 2.a.1. Number and Type of infrastructures <br> 2.a.2. Access policies in place <br><br> 2.b.1. Number of services offered <br> 2.b.2. Enrolled services <br><br> 2.c.1. National (regional) registry or other federation mechanisms for data in place/planned <br> 2.b.6. FAIR-enabling services | Architecture indicators National (regional) registry or other federation mechanisms for data in place/planned National (regional) dataset catalogue(s) in place/planned National PID Policy User satisfaction Data usage Interoperability API usage |
| Undeveloped training for Open Science (landscaping data) | Training and skills resources created <br><br> Training and skills and programmes implemented | C. Training & Skills <br> 3.a. Community Metrics <br> 3.a.1 National/ regional curricula in place/planned <br> 3.a.2. Basic training available for researchers and research support staff (e.g., National Competence centres <br> 3.a.3. Number of trained people per year <br> 3.a.4. Number of training material | National/regional curricula in place/planned <br><br> Basic training available for researchers and research support staff <br><br> People trained per year |
| Linguistic diversity | Relevant information is made available in local languages | N/A because NOSCI indicators are applied at the national level, while this is a regional challenge | N/A |

The following passages briefly summarize the status of the scholcomm-related indicators in SEE. Although a progress from the initial stage is obvious for all identified challenges, some issues cannot be resolved at the regional level and the final success depends on the actions taken by national initiatives [24].

The establishment of 15 NOSCIs in SEE, one of NI4OS-Europe project outputs, is characterized by multistakeholder governance models and forms, such as task forces, consortia, and national projects. Their role is considered prominent concerning the development of open science ecosystems, especially the EOSC vision, and they consequently have a remarkable impact in the research environment and scholarly communication.

In this context, the formation of national and institutional OS policies becomes a priority. Interested stakeholders join forces to manage OS as a national issue and agree on a common framework by signing a Memorandum of Understanding (MoU). The newly developed national and institutional OS policies explicitly address various aspects of OS, also encompassing incentives for open science activities.

The non-EU countries are encouraged and decisively supported to extend their participation in OS initiatives and networks. The adoption of best practices recommended by NI4OS-Europe improves their scholarly environment and makes them achieve compliance

with the EOSC Rules of Participation. In particular, the partner countries are encouraged to join the EOSC Association.

Furthermore, in order to promote capacity building for scholarly communication, OS policies address the issue of raising funds from sources of various levels. The ultimate goal is thus to absorb a share from the national budget and to take advantage of all possible opportunities in European and international funding programs. It is important to note that partner countries, including non-EU members, are now included in Europe's strategic plans and Horizon Europe's Working Programs (e.g., European Commission, Directorate-General for Research and Innovation, Strategic foresight in the Western Balkans: recovery on the horizon, Publications Office of the European Union, 2021, https://data.europa.eu/doi/10.2777/202437 and European Commission, Directorate-General for Research and Innovation, Commission Implementing Decision C (2022)2975, Horizon Europe Work Programme 2021–2022. Widening participation and strengthening the European Research Area, 10 May 2022. https://eur-lex.europa.eu/resource.html?uri=cellar:c1f95e49-d11b-11ec-a95f-01aa75ed71a1.0001.02/DOC_12&format=PDF, accessed on 30 June 2022) in the area of Research and Development, aiming specifically at promoting the transition to a new research framework for scholarly communication and the adoption of new innovative practices including OS.

The local and regional infrastructure and access to it have been improved through the effort to prepare services in the NI4OS-Europe partner countries for onboarding to the EOSC Catalogue of Services and Marketplace. As an intermediate step in this process, the NI4OS-Europe Service Catalogue (https://catalogue.ni4os.eu/, accessed on 30 August 2022) has been established [24]. It provides information about selected repositories, thematic, generic and core services in the partner countries, as well as about service policies. User manuals and training materials are provided, as well as a helpdesk. The services are monitored, and the monitoring information is publicly available [25]. A set of tools have been developed to support FAIR and Open Research Data Management and inclusion to EOSC: the Licence Clearance Tool (LCT) to automate license clearance; the EOSC Rules of Participation Tool (RoLECT) specifically addressing legal and ethical compliance for EOSC; the Repository Policy Generator (RePol), a tool for drafting repository and privacy policies. The project has created robust guidelines and a network of experts to support the process of establishing national service catalogues. The visibility of local and regional services has improved significantly. At the same time, the Catalogue makes it easier for research communities to find reliable tools and services.

Progress has also been made in terms of the limited availability of information on OS policy and activities across the region, through the launch of new informative resources and the creation of new materials. All interested parties may consult stakeholder registries, such as the Stakeholder Map in NI4OS-Europe website to trace local and regional actors. In addition, they may access service registries—among them the NI4OS-Europe Service Catalogue to find more about services in the region and their policies. Moreover, they now have at their disposal policy registries so as to become aware of what applies in each country together with information about infrastructure. The latter are hosted mainly in the NOSCIs' pages and the emerging NOSCIs' portals.

At the same time, training for researchers and research support staff is organized, specialized material and skills resources are produced, covering OS practices in several aspects of scholarly communication. The training materials created during the project are available on the NI4OS-Europe Training Platform [23]. This action increases consciousness regarding the potential of OS and promotes the use of the available resources for the benefit of research and scientific knowledge in the countries and worldwide.

Finally, it is worth mentioning that there is an overall approach to overcome limitations due to language diversity, by providing informative and training material not only in English but also translated in local languages of the partner countries.

The NOSCIs' approach has been very successful and already 10 NOSCIs have been established while 5 more are on the course to be established. The concept of NOSCI is

flexible and it allows for various governance models and stakeholder involvement. In the NI4OS-Europe countries where NOSCIs are established, they already play a prominent role in the facilitation of the EOSC governance and also as enablers of EOSC inclusion and Open Science widening at the national level. They are thus directly involved or are even leading scholarly communication activities in their countries. Relevant examples include the contributions to the drafting of national OS plans, the successful implementation of institutional OS strategies, the organisation of dissemination events on OS, the delivery of training events and material in relation to Open Research Data Management (ORDM).

## 6. Conclusions

The NI4OS-Europe project supports the development of OS and FAIR policies in 15 Member States and Associated Countries as well as the development and inclusion of the national Open Science Cloud Initiatives (NOSCIs) in the overall scheme of EOSC governance. By doing so, it increases engagement within a trusted, federated environment that allows researchers to search, reuse and publish data and services and builds OS capacity in the region, contributing directly to fundamental priorities for scholarly communities.

To amplify the reliability of the approach, we considered it important to provide an evaluation methodology for the process of establishing, operating and monitoring the results of the NOSCIs. The NOSCI indicators have been re-examined in relation to their suitability in the scholarly communication context, and a set of relevant metrics has been derived, along with the challenges to which they respond.

Considering the higher degree of complexity in our region (political, historical, OS policies) we believe that the set of indicators for scholarly communication in relation to EOSC/OS activities has a high reproducibility potential. The solutions offered by the NI4OS-Europe project are flexible and adaptable. In this respect, this analysis may be useful not only to Open Science stakeholders in Europe but also in other parts of the world, particularly in developing countries.

**Author Contributions:** Conceptualization, M.Š. and E.T.; methodology, M.Š. and E.T.; validation, E.S., K.K. and E.T.; formal analysis, E.S., K.K. and M.Š.; investigation, B.K., E.P., E.S., K.K., I.P. and K.L.; data curation, E.S. and K.K.; writing—original draft preparation, M.Š., E.T., K.L. and I.P.; writing—review and editing, E.S., K.K. and E.T.; visualization, M.Š., I.P., E.S. and K.K.; supervision, E.T.; project administration, E.T.; funding acquisition, E.T. All authors have read and agreed to the published version of the manuscript.

**Funding:** This research was funded by the European Commission, under the Horizon 2020 European research infrastructures, grant number 857645.

**Data Availability Statement:** No original, raw data has been generated for this manuscript. The used data sources are listed under Data and Methodology.

**Acknowledgments:** The authors would like to thank all partners in the NI4OS-Europe project for their help with data collection throughout the project.

**Conflicts of Interest:** The authors declare no conflict of interest.

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
