# Peer review of "Building National Open Science Cloud Initiatives (NOSCIs) in Southeast Europe: Supporting Research and Scholarly Communication"

_publications, doi:10.3390/publications10040042_

Round 1
Reviewer 1 Report
Review of ‘Building National Open Science Cloud Initiatives (NOSCIs) in Southeast Europe: supporting research and scholarly communication’
1. Title, Abstract and Keywords are fine.
2. The paper presents the results of a project with European funding. It is well written and detailed, with several illustrations showing the results.
3. I think that Data and Methodology Section should improve the links to the original results, i.e. Reference 9 has no mentioned link. That sort of connection would allow the reader to gain access to the original research efforts.
4. Numbering of the sections should be corrected, starting with Line 455 until the end of the paper.
Author Response
Dear reviewer,
thank you very much for your comments! Please find below our reply to your relevant comments. Changes have been marked in red in the document.
Best regards,
NI4OS-Europe team
- I think that Data and Methodology Section should improve the links to the original results, i.e. Reference 9 has no mentioned link. That sort of connection would allow the reader to gain access to the original research efforts.
-->The DOI assigned to the dataset was added to Reference 9, as well as a reference to the World Bank Data.
- Numbering of the sections should be corrected, starting with Line 455 until the end of the paper.
-->Has been corrected
Reviewer 2 Report
This paper points out the many challenges of building open research infrastructure that spans and integrates multiple countries, particularly those with different languages, struggling economies, and political instabilities. It identifies some potential solutions, or at least mitigating factors, that have arisen through the establishment of the EOSC and the NOSCIs. This is useful information for other countries, particularly developing nations, who wish to participate in the global open science movement to leverage access to start-of-the-art computation and data services.
I found a lot of the introductory material to be rather obvious and I think it could probably be reduced by ~50% without compromising the value of the paper.
The survey data analysis is a bit superficial, but perhaps necessarily so because the authors point out on several occasions some of the deficiencies in the data. Some of the tables and figures could be a bit better annotated, as not all of the acronyms and abbreviations will be familiar to all readers.
I also found Table 5 a bit superficial, where each challenge was simply refuted by its obvious solution. "Lack of incentives..." / "Incentives are included...", "Uneven infrastructure..." / "Improved infrastructure...". Could this information be compressed into a simple list of goals with the corresponding NI4OS and EOSC enablers?
The authors might find it interesting to explore the NIST Research Data Framework as a tool for assessing capabilities and competencies in research data management. https://www.nist.gov/programs-projects/research-data-framework-rdaf
Author Response
Dear reviewer,
thank you very much for your comments! Please find below our reply to your relevant comments. Changes have been marked in red in the document.
Best regards,
NI4OS-Europe team
========
I found a lot of the introductory material to be rather obvious and I think it could probably be reduced by ~50% without compromising the value of the paper.
-->The introduction was reduced.
===
The survey data analysis is a bit superficial, but perhaps necessarily so because the authors point out on several occasions some of the deficiencies in the data. Some of the tables and figures could be a bit better annotated, as not all of the acronyms and abbreviations will be familiar to all readers.
--> Acronyms are resolved, either in the text or in footnotes.
===
I also found Table 5 a bit superficial, where each challenge was simply refuted by its obvious solution. "Lack of incentives..." / "Incentives are included...", "Uneven infrastructure..." / "Improved infrastructure...". Could this information be compressed into a simple list of goals with the corresponding NI4OS and EOSC enablers?
-->The purpose of Table 5 is to show the applicability of the indicators developed by the NI4OS-Europe project (Blueprint metrics) in the context of scholarly communication. The Blueprint metrics have been developed for a more general purpose. In this manuscript, an attempt is made to derive from the general Blueprint metrics a set of indicators that could be used for impact assessment in the context of scholarly communication. Therefore, the Indicators column in Table 5 does not list obvious solutions to the identified problems but the indicators of success in coping with the challenges in the area of scholarly communication. This is not an exhaustive list, as the analysis is limited to Southeast Europe and only the regional level is discussed. However, we believe that the proposed set of indicators could be tested in other environments and amended. Keeping in mind that Table 5 shows the genesis of this set of indicators, we find it important to retain the table as it is. At the same time, we provide additional explanation of its structure in the text.
===
The authors might find it interesting to explore the NIST Research Data Framework as a tool for assessing capabilities and competencies in research data management. https://www.nist.gov/programs-projects/research-data-framework-rdaf
-->Thank you for the suggestion. This tool will certainly be very useful in our further work and it is especially relevant for the national initiatives. In this manuscript, we sought to keep the discussion at the regional level.